# Acetate Production by *Moorella thermoacetica* via Syngas Fermentation: Effect of Yeast Extract and Syngas Composition

Budi Mandra Harahap [1,2] and Birgitte K. Ahring [1,2,3,*]

1 Bioproducts, Science, and Engineering Laboratory, Washington State University Tri-Cities, 2710, Crimson Way, Richland, WA 99354, USA; budi.harahap@wsu.edu
2 Department of Biological System Engineering, Washington State University, L. J. Smith Hall, Pullman, WA 99164, USA
3 Voiland School of Chemical Engineering and Bioengineering, Washington State University, Wegner Hall, Pullman, WA 99164, USA
* Correspondence: bka@wsu.edu

**Abstract:** Gasifiers produce a gaseous mixture of $CO/CO_2/H_2$, also known as synthesis gas (syngas), containing varying compositions and ratios depending on the lignocellulose material types, gasifier design, and gasification conditions. Different physicochemical and thermodynamic properties of each gas type in the various syngas blends can influence syngas fermentation performance for the production of chemicals such as acetate. This study examined the effect of syngas composition (CO, $CO/H_2$, $CO/CO_2/H_2$, and $CO/H_2$) and its corresponding ratio on acetate production using *Moorella thermoacetica*, a thermophilic homoacetogen as the biocatalyst. We also investigated the effect of yeast extract addition for enhancing acetate production. A syngas fermentation study performed at a total pressure of 19 psig (2.29 atm) demonstrated that syngas fermentation in the absence of CO ($30\%CO_2/70\%H_2$) or at low CO proportions ($21\%CO/24\%CO_2/55\%H_2$) resulted in the highest volumetric productivity of acetate ($0.046 \pm 0.001$ and $0.037 \pm 0.001$ g/L/h, respectively). Interestingly, syngas fermentation without CO reached the highest $Y_{P/X}$ of $22.461 \pm 0.574$ g-acetate/g-biomass, indicating that more acetate was produced compared to cell biomass. Higher biomass production was obtained when the CO proportion was increased up to 75% in $CO/H_2$ fermentation. However, the cell growth and acetate production dramatically decreased with increasing CO proportion up to 99.5% CO as the sole constituent of the syngas. Even so, acetate production using 99.5% CO could be improved by adding 2 g/L yeast extract.

**Keywords:** acetate production; syngas fermentation; gas compositions; thermophilic; homoacetogen; *Moorella thermoacetica*





## 1. Introduction

Growing attention to the application of carbon-neutral methods for producing chemicals has shifted the paradigm to using renewable biomass feedstock instead of fast-depleting fossil fuel with minimum waste dumped. In lignocellulose biorefinery, lignin, the second major polymer comprising 15–30% (dry weight) of the total lignocellulosic biomass [1], becomes a leftover component dissolved into a spent liquor after pretreatment (delignification) and ending as a waste product [2]. Gasification of lignocellulosic biomass materials followed by syngas fermentation offers a solution for using lignin as a raw material instead of wasting this part. Fermentation is further operated under moderate conditions whenever the syngas is produced from gasification [3]. Apart from bioethanol, acetate is a potential chemical that can be produced through the syngas route. Acetate is the ester form of acetic acid and has enormous worldwide market demand [4] with a broad spectrum of applications such as building block for syntheses to vinyl acetate monomer (VAM) [5], acetate anhydrate [6], acetate esters [7], monochloroacetic acid [8], and also as a solvent to produce dimethyl terephthalate and terephthalic acid [9].

Syngas is a gaseous mixture of CO, $CO_2$, and $H_2$ thermochemically generated from lignocellulose gasification [10]. This gaseous blend can be a substrate for homoacetogens to synthesize acetate through the Wood–Ljungdahl pathway (WLP) or the reductive acetyl-coenzyme A (Acetyl-CoA) pathway [11–13]. The microorganism that has the capability of metabolizing syngas through WLP is carboxydotrophic homoacetogens [14]. More specifically, carboxydotrophic homoacetogen is autotrophic-acetogen-resistant to high CO concentrations. Compared to the other carboxydotrophic homoacetogens commercially available, *Moorella thermoacetica* is a thermophilic bacterium with a high growth rate for syngas-to-acetate conversion with acetate as a sole product, unlike other carboxydotrophic homoacetogens also synthesizing other organic acids and alcohols [15,16]. Additionally, thermophilic syngas fermentation is preferred due to the low energy requirement for gas cooling and higher fermentation rate than organisms growing under mesophilic conditions [17].

*M. thermoacetica* utilizes CO as both a carbon source and an electron donor for acetate production through WLP [3,16]. In WLP, carbon monoxide dehydrogenase (CODH) catalyzes CO oxidation by releasing 2 mol electrons per 1 mol CO using ferredoxin or NADH as an electron carrier to form $CO_2$ [15,18,19]. The formed $CO_2$ is then converted into a methyl group compound in the eastern branch of WLP or reduced back into CO in the western branch of WLP. The supplied CO or in situ generated CO from $CO_2$ reduction is condensed with CoA and a methyl group compound into acetyl CoA, catalyzed by acetyl CoA synthase (ACS). The acetyl CoA is further catalyzed by phosphotransacetylase (PTA) and acetate kinase (ACK) to form acetate by releasing ATP. Therefore, according to this metabolic pathway, it is possible to ferment CO as a sole gas using homoacetogens. The stoichiometric reaction of CO-to-acetate conversion called the biological water–gas shift reaction (BWGSR), follows the equation below [20,21]:

$$4 \, CO + 2 \, H_2O \rightarrow CH_3COOH + 2 \, CO_2 \quad \Delta G° = -154.6 \, kJ/mol. \quad (1)$$

According to Equation (1), the CO fermentation through BWGSR is thermodynamically favorable and stoichiometrically releases 2 mol $CO_2$ per 1 mol acetate. Since the formed $CO_2$ must further be converted into acetate for complete syngas conversion, homoacetogens require another electron donor besides CO to reduce the formed $CO_2$ into acetate. The electron source can be obtained from $H_2$, another gas produced during gasification. For complete CO conversion, 4 mol $H_2$ is required to convert 2 mol $CO_2$ from CO fermentation, as shown in Equation (2). The reaction of CO and $H_2$ can be seen in Equation (3).

$$4 \, H_2 + 2 \, CO_2 \rightarrow CH_3COOH + 2 \, H_2O \quad \Delta G° = -74.3 \, kJ/mol \quad (2)$$

$$2 \, CO + 2 \, H_2 \rightarrow CH_3COOH \quad \Delta G° = -114.5 \, kJ/mol \quad (3)$$

One of the problems of syngas fermentation is various syngas compositions out of the gasifier off-gas depending on biomass properties, gasifier type and design, and gasification conditions [22]. The ratio of $H_2$ and CO ranges from 0.2 to 4.6, whereas the typical portion of $CO_2$ is in the range of 5–15% [23]. These various syngas compositions and ratios could affect fermentation performance due to the different physicochemical and thermodynamic properties of each gas in the syngas blend as well as the gas toxicity effect, ultimately causing lower acetate production from the fermentation compared to theoretical results.

For example, a higher CO portion in the syngas blends led to more CO accumulation in the fermentation broth. Because homoacetogen only tolerates CO up to a certain concentration [24], too high of a CO proportion might lead to cell growth inhibition and low acetate formation. On the other hand, the thermodynamic analysis revealed that electron generation from CO is thermodynamically more favorable than that from $H_2$, independent of pH, ionic strength, gas partial pressure, and electron carrier pairs [25]. This indicates that CO is a preferred electron source over $H_2$, which can generate more electrons from CO at a higher rate. Furthermore, CO was an inhibitor for hydrogenase, an enzyme catalyzing $H_2$

oxidation for electron generation, leading to more electrons produced and less $H_2$ utilized with increasing CO proportion in the syngas mixture [26,27]. This indicated that $CO/H_2$ and its ratio hypothetically might influence syngas fermentation via *M. thermoacetica*, which would be tested in this study.

In the case of $CO_2/H_2$ fermentation, $H_2$ is required at a sufficient amount in syngas blends for the complete conversion of $CO_2$. However, the bottleneck of $CO_2/H_2$ fermentation is the low solubility of $H_2$ in the fermentation broth [28]. Based on the physicochemical properties, $H_2$ has the lowest Henry's law constant, indicating the least solubility, while $CO_2$ is the most soluble gas in the water over the other syngas components [29]. By increasing $H_2$ concentration correlated to the increase in $H_2$ partial pressure, the mass transfer limitation issue of $H_2$ will be mitigated. Another study also mentioned that a low $CO/CO_2$ ratio reduced cell growth, confirming that $CO_2$ inhibited CO oxidation [30]. Therefore, the specific ratios of $CO/CO_2/H_2$ must be evaluated to find the conditions for maximum acetate production.

Understanding how the syngas composition influences syngas fermentation performance is required to obtain an efficient process with high syngas conversion rates. Several previous studies only reported acetate production using various strains of *M. thermoacetica* using CO [31–33], $CO_2/H_2$ [31–34], $CO/H_2/CO_2$ [35], and $CO/CO_2$. No previous study has dealt with the effect of syngas composition on acetate production, specifically for this bacterium. Most of the previous works have further focused on bioethanol production [26,36–38]. Thus, the first objective of the present study is to investigate the impact of syngas composition on acetate production using *M. thermoacetica*.

The presence of CO during syngas fermentation negatively affects cell growth and acetate formation. Studies conducted by Benevenuti et al. [39] and Hongrae et al. [40] reported that yeast extract (nitrogen and vitamin-rich nutrient) could increase cell growth of *Clostridium carboxidivorans* and *Clostridium autoethanogenum* for bioethanol production using syngas. However, the problem of yeast extract for syngas fermentation is that this nutrient largely contributed to the total medium cost [41–44]. Hence, in this study, we also examined the effect of yeast extract addition on acetate production from CO and total medium cost for syngas fermentation using *M. thermoacetica*.

## 2. Materials and Methods

### 2.1. Strain and Inoculum Preparation

The strain of *Moorella thermocetica* DSM 2955 was from DSMZ (Deutsche Sammlung von Mikroorganismen und Zellkulturen)—German Collection of Microorganisms. This strain was grown in DSM medium 316 at 55 °C and then stored at 4 °C for further experiments. Prior to fermentation, the inoculum was prepared in BA media [45,46]. The BA media and vitamin solution composition can be seen in Table 1. The inoculum was incubated in a rotary shaker at 55 °C, 210 rpm for 48 h.

### 2.2. Batch Syngas Fermentation

In this study, 10% of the *M. thermoacetica* inoculum was aseptically and anaerobically inoculated into 160 mL serum vial containing 45 mL sterile BA media, including filter-sterilized vitamin solution with or without 2 g/L yeast extract addition. The headspace of serum vial was then flushed and pressurized with syngas up to 19 psig using various gas composition and ratio involving 99.5% CO, 25%CO/75%$H_2$, 50%CO/50%$H_2$, 75%CO/25%$H_2$, 21%CO/24%$CO_2$/55%$H_2$, 41%CO/18%$CO_2$/41%$H_2$, 61%CO/12%$CO_2$/27%$H_2$, and 30%$CO_2$/70%$H_2$. Fermentation was performed in a rotary shaker at 55 °C with stirring at 210 rpm. The pressure was periodically analyzed using a pressure gauge, and the fermentation liquid was sampled everyday for 7 days to measure cell and acetate concentration.

**Table 1.** Basic Anaerobic (BA) media composition and cost analysis.

| Components | Concentration [mg/L] | Chemical Price [$/g] * | Cost [$/L-Solution] | % of Total Cost |
|---|---|---|---|---|
| Yeast Extract | 2000 | 0.281 | 0.562 | 38.959 |
| NH$_4$Cl | 1000 | 0.081 | 0.081 | 5.601 |
| NaCl | 100 | 0.079 | 0.008 | 0.547 |
| MgCl$_2$.6H$_2$O | 100 | 0.199 | 0.020 | 1.380 |
| CaCl$_2$.2H$_2$O | 50 | 0.162 | 0.008 | 0.562 |
| K$_2$HPO$_4$.3H$_2$O | 400 | 0.313 | 0.125 | 8.679 |
| Resazurin sodium salt | 0.5 | 0.038 | 0.019 | 1.317 |
| Trace Elements: | | | | |
| H3BO3 | 0.05 | 0.200 | 0.00001 | 0.001 |
| ZnCl2 | 0.05 | 0.230 | 0.00001 | 0.001 |
| CuCl2.2H2O | 0.038 | 0.550 | 0.00002 | 0.001 |
| MnCl2.2H2O | 0.041 | 0.360 | 0.00001 | 0.001 |
| (NH4)6Mo7O24.4H2O | 0.05 | 0.450 | 0.00002 | 0.002 |
| AlCl3.6H2O | 0.09 | 0.166 | 0.00001 | 0.001 |
| CoCl2.6H2O | 0.05 | 0.566 | 0.00003 | 0.002 |
| NiCl2.6H2O | 0.092 | 0.266 | 0.00002 | 0.002 |
| Na2EDTA.2H2O | 0.5 | 0.470 | 0.00024 | 0.016 |
| Na2SeO3.5H2O | 0.1 | 2.363 | 0.00024 | 0.016 |
| Total | | | 0.001 | 0.040 |
| Sodium Bicarbonate | 2600 | 0.103 | 0.268 | 18.564 |
| Vitamins: | | | | |
| Biotin | 0.02 | 90.9 | 0.00182 | 0.126 |
| folic acid | 0.02 | 5.64 | 0.00011 | 0.008 |
| pyridoxine hydrochloride | 0.1 | 22.1 | 0.00221 | 0.153 |
| riboflavin | 0.05 | 4.49 | 0.00022 | 0.016 |
| thiamine hydrochloride | 0.05 | 3.92 | 0.00020 | 0.014 |
| cyanocobalamin | 0.001 | 122 | 0.00012 | 0.008 |
| nicotinic acid | 0.05 | 0.446 | 0.00002 | 0.002 |
| p-aminobenzoic acid | 0.05 | 5.06 | 0.00025 | 0.018 |
| thioctic acid | 0.05 | 21.6 | 0.00108 | 0.075 |
| DL-pantothenic acid | 0.05 | 7.2 | 0.00036 | 0.025 |
| Total | | | 0.006 | 0.44 |
| L-Cysteine | 0.47 | 0.74 | 0.345 | 23.905 |
| Total Cost | | | 1.442 | 100 |

* Prices of chemical was from Sigma-Aldrich (August 2023).

### 2.3. Analytical Procedures

The optical density (OD) of the samples was analyzed using a Thermo Scientific™ Genesys™ 150 visible-UV spectrophotometer at 600 nm wavelength. The dry cell weight was counted based on the prepared standard curve (DCW = 0.428 OD). The acetate concentration in the fermentation broth was measured using a Dionex UltiMate 3000 high-performance liquid chromatography (HPLC) (Sunnyvale, CA, USA) with BioRAD Aminex 87-H column (Hercules, CA, USA) and a Shodex RI-101 refractive index detector (New York, NY, USA). The operational condition of HPLC followed the procedure developed by Garret et al. [45]. Syngas composition was measured using a universal gas analyzer (UGA series, Standford Research System, Sunnyvale, CA, USA).

### 2.4. Calculation and Statistical Analysis

The maximum specific growth rate (μ) in 1/h was obtained during the exponential phase and calculated following Equation (4):

$$\mu_{max} = \frac{Ln\ (X_f \times V_f) - Ln\ (X_i \times V_i)}{t_f - t_i}. \tag{4}$$

$X_i$ and $X_f$ (g/L) denoted the biomass concentration at the initial exponential phase ($t_i$) and the end of the exponential phase ($t_f$) in h, respectively, and $V_i$ and $V_f$ (L) represented the initial and the final working volume at the exponential phase, respectively. The acetate volumetric productivity ($Q_P$) in g/(L.h) and acetate yield ($Y_{P/X}$) in g/g were counted following the Equations (5) and (6).

$$Q_P = \frac{(P_f \times V_f) - (P_i \times V_i)}{V_f - t_f} \tag{5}$$

$$Y_{P/X} = \frac{(P_f \times V_f) - (P_i \times V_i)}{(X_f \times V_f) - (X_i \times V_i)} \tag{6}$$

$P_f$ and $P_i$ (g/L) referred to the acetate concentration at the initial exponential phase and the end of the exponential phase, respectively. Carbon proportion (%) of each component and carbon conversion efficiency of acetate or CCE (%) were calculated following the Equations (7) and (8).

$$Carbon\ proportion = \frac{Carbon\ mass\ of\ component\ i\ [g]}{Total\ carbon\ mass\ [g]} \times 100\% \tag{7}$$

$$CCE\ of\ Acetate = \frac{Carbon\ mass\ of\ the\ produced\ acetate\ [g]}{Total\ carbon\ mass\ of\ the\ consumed\ syngas\ [g]} \times 100\% \tag{8}$$

In this study, the statistical analysis encompassed two-sample *t*-tests to determine whether the means of two groups were statistically different, while analysis of variance (ANOVA) was used for more than two groups. This study used a 95% confidence interval. After the ANOVA test, a post hoc test (Tukey's test) was conducted and the data were computed using R programming language.

## 3. Results

### 3.1. CO Fermentation and Yeast Extract Effect

Figure 1 depicts cell growth and acetate formation profile over time during CO fermentation in the presence and the absence of yeast extract. The pressure drop displayed in Figure 1 represents CO consumption. According to the cell growth profile of *M. thermoacetica* (Figure 1), the exponential phase started after 100 h of the adaptation phase in CO fermentation without yeast extract addition. The exponential phase lasted for 96 h with a rate of $0.02 \pm 0.00$/h and a maximum dry cell weight of $0.60 \pm 0.03$ g/L. Further observation revealed that 2 g/L yeast extract addition slightly reduced the lag phase duration and resulted in a 1.8 and two-fold increase in the maximum specific growth rate ($\mu_{max}$) and cell production (Figure 2), respectively.

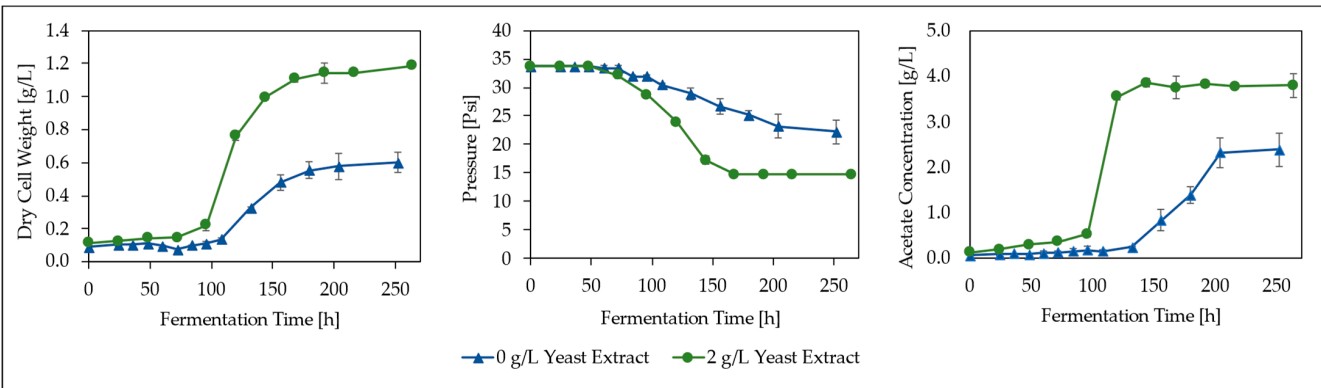

**Figure 1.** The profile of cell growth, gas pressure, and acetate concentration during syngas fermentation with 2 g/L yeast extract and in the absence of yeast extract.

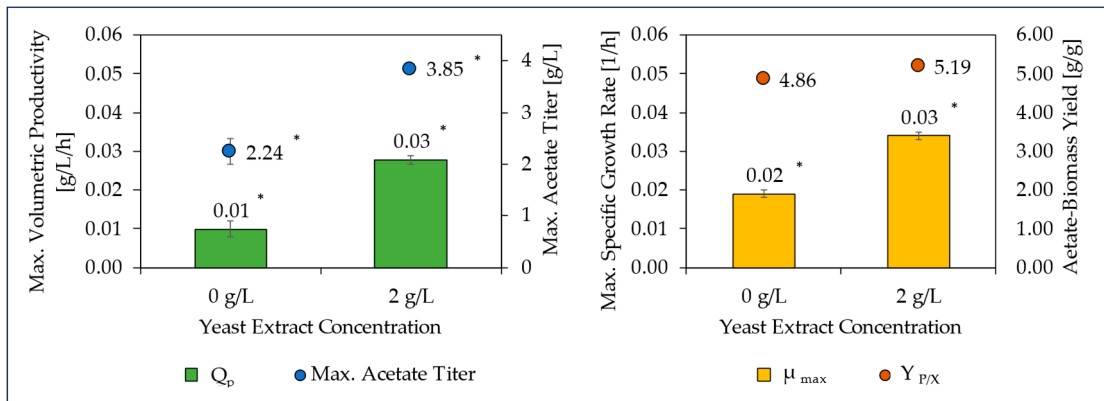

**Figure 2.** Maximum specific growth rate, acetate–biomass yield, volumetric productivity, and acetate titer obtained from 99.5% CO fermentation with or without 2 g/L yeast extract. An asterisk (*) represents a statistically significant difference between syngas fermentation with and without 2 g/L yeast extract for each parameter.

Besides positively affecting *M. thermoacetica* cell growth, yeast extract addition improved acetate production titer and productivity. In CO fermentation with no yeast extract added, *M. thermoacetica* only produced 2.24 ± 0.24 g/L acetate with low productivity (0.01 ± 0.00 g/L/h) as shown in Figure 2. The acetate production enhanced up to 3.85 ± 0.09 g/L by adding 2 g/L yeast extract. The volumetric productivity of acetate also increased 2.5-fold due to the effect of yeast extract addition.

Acetate production enhancement by adding yeast extract was associated with more gas consumption at the end of CO fermentation. Even though yeast extract will enhance acetate production, larger-scale acetate production needs a less costly fermentation media without yeast extract. Cost analysis of BA media revealed that yeast extract contributed 39% of the total medium cost, followed by L-cysteine (23.9%) and sodium bicarbonate (18.6%), while the rest of the nutrients only accounted for 10% of the cost. A tiny portion of the cost of the media was vitamins and trace elements (below 1%). Besides the addition of yeast extract during CO fermentation, acetate production can be improved by adjusting the syngas composition.

*3.2. Effect of Syngas Composition*

The objective of this experiment is to examine the effect of syngas composition on cell growth, gas consumption, and acetate formation. The results are presented in Figure 3a (only CO), Figure 3b (CO/$H_2$), Figure 3c (CO/$H_2$/$CO_2$), and Figure 3d ($CO_2$/$H_2$). Overall, the growth kinetics of *M. thermoacetica* were associated with acetate formation and syngas

consumption, where cells and acetate were formed due to syngas consumption. However, several parameters, such as the adaptation and exponential phase duration, maximum cell growth rate, cell production, acetate titer, productivity, and remaining substrate composition, were different during different syngas fermentations. In Figure 3a, fermentation using 99.5% CO needed a prolonged adaptation phase of above 100 compared to other syngas compositions. The adaptation phase significantly reduced up to less than 100 h when the CO proportion was reduced from 50% in $CO/H_2$ blend (Figure 3b) and was only 24 h at 41% CO in a $CO/CO_2/H_2$ mixture (Figure 3c). The lowest lag phase of below 24 h was reached when CO was removed from the syngas blend or without CO, such as using $CO_2/H_2$ (Figure 3d). This observation indicated that CO at a high percentage in the syngas needed more adaptation than at lower CO concentrations.

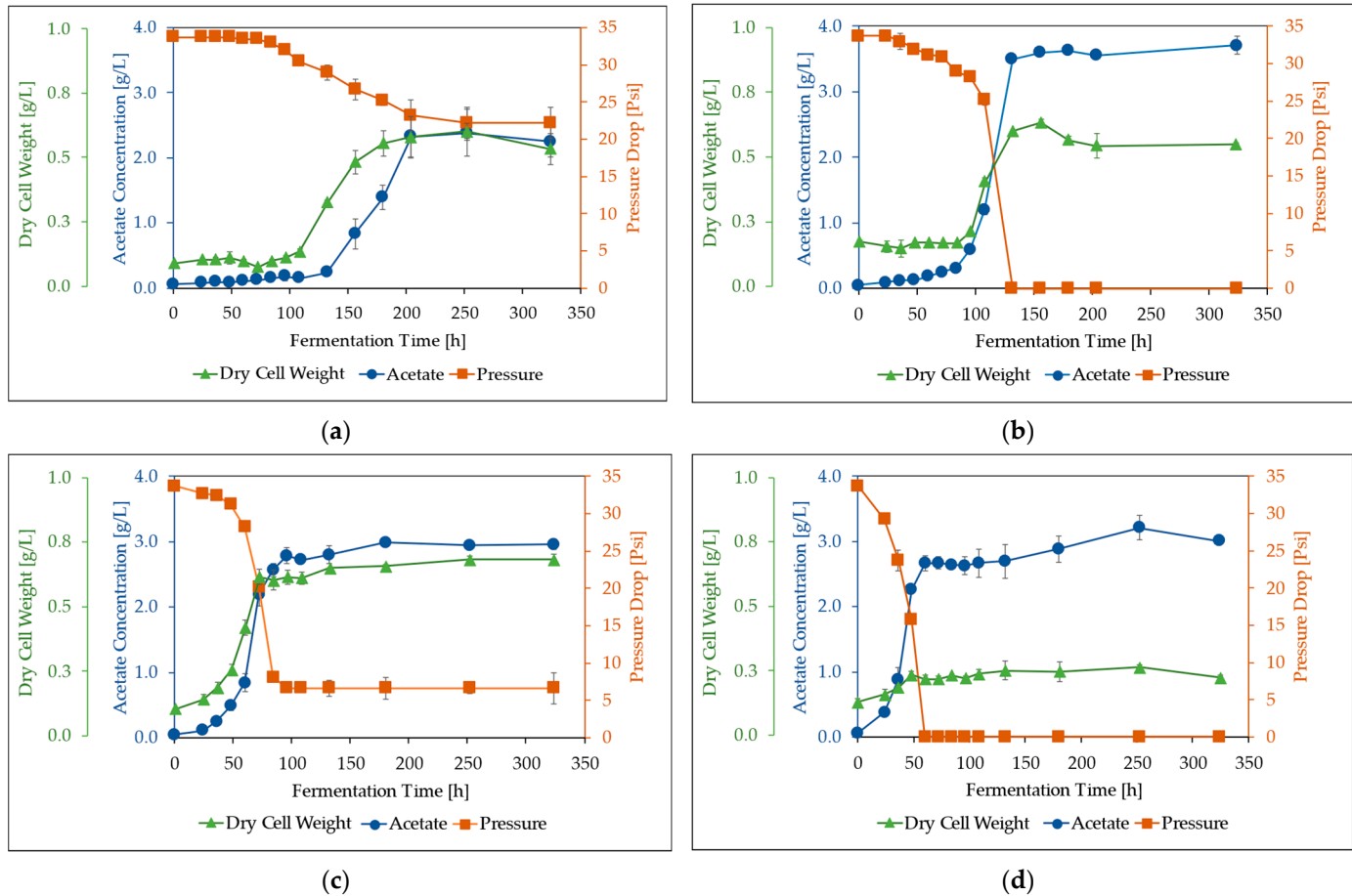

**Figure 3.** The profile of cell growth, gas pressure drops, and acetate concentration during 99.5%CO (**a**), 50%CO/50%$H_2$ (**b**), 41%CO/18%$CO_2$/41%$H_2$ (**c**), and 30%$CO_2$/70%$H_2$ (**d**) fermentation in the absence of yeast extract addition.

As shown in Table 2, CO presence in the syngas mixture significantly affected the specific growth rate (*p*-value < 0.05). In syngas fermentation without yeast extract, the composition of 99.5% CO showed a low specific growth rate of 0.02 ± 0.00/h, compared to 50%CO/50%$H_2$ (0.04 ± 0.00/h) and 41%CO/18%$CO_2$/41%$H_2$ (0.03 ± 0.00/h). By adding 2 g/L yeast extract, the specific growth rate in 99.5%CO fermentation was statistically similar to 30%CO/70%$H_2$ fermentation (*p*-value > 0.05). The lowest specific growth rate (only 0.02 ± 0.00/h) and biomass production (Figure 3d) were observed in the absence of CO in the syngas blend, such as 30%$CO_2$/70%$H_2$ fermentation. This interesting observation clearly showed that the CO presence at a certain proportion in the syngas blend allowed *M. thermoacetica* to produce more cells with an accelerated cell growth rate, while high

CO concentrations, such as 99.5%CO, led to inhibition of the specific growth rate and cell production.

**Table 2.** Maximum specific growth rate, specific acetate production rate, volumetric productivity, and acetate titer obtained from syngas fermentation at various syngas compositions and ratios. The different superscript letters showed that the mean among groups was significantly different (*p*-value < 0.05).

| Syngas Composition | Parameters | | | |
|---|---|---|---|---|
| | $\mu_{max}$ [1/h] | $Y_{P/X}$ (g/g) | $Q_{P\,max}$ [g/L/h] | Max. Acetate Titer [g/L] |
| 99.5%CO without yeast extract | 0.02 ± 0.00 [a] | 4.86 ± 0.12 [a] | 0.01 ± 0.00 [a] | 2.38 ± 0.36 [a] |
| 99.5%CO with 2 g/L yeast extract | 0.03 ± 0.00 [b] | 5.19 ± 0.05 [a] | 0.03 ± 0.00 [b] | 3.85 ± 0.09 [b] |
| 30%$CO_2$/70%$H_2$ | 0.02 ± 0.00 [c] | 22.46 ± 0.57 [b] | 0.05 ± 0.00 [c] | 3.21 ± 0.18 [b] |
| 25%CO/75%$H_2$ | 0.02 ± 0.00 [d,e,f] | 9.54 ± 0.00 [c] | 0.02 ± 0.00 [d] | 3.44 ± 0.25 [b] |
| 50%CO/50%$H_2$ | 0.04 ± 0.00 [b] | 7.56 ± 0.14 [d] | 0.03 ± 0.00 [e,f] | 3.71 ± 0.13 [b] |
| 75%CO/25%$H_2$ | 0.04 ± 0.00 [b] | 4.62 ± 0.03 [a] | 0.02 ± 0.00 [d] | 3.35 ± 0.17 [b] |
| 21%CO/24%$CO_2$/55%$H_2$ | 0.02 ± 0.00 [d] | 6.38 ± 0.16 [e] | 0.04 ± 0.00 [g] | 3.36 ± 0.16 [b] |
| 41%CO/18%$CO_2$/41%$H_2$ | 0.03 ± 0.00 [d] | 4.30 ± 0.10 [a] | 0.03 ± 0.00 [h] | 2.99 ± 0.01 [a,b] |
| 61%CO/12%$CO_2$/27%$H_2$ | 0.03 ± 0.00 [d] | 4.84 ± 0.57 [a] | 0.03 ± 0.00 [f,h] | 3.05 ± 0.06 [a,b] |

Although $CO_2$/$H_2$ fermentation had the lowest cell and specific growth rate, the acetate production was statistically the same as CO fermentation using yeast extract, CO/$H_2$, or CO/$CO_2$/$H_2$ fermentation (*p*-value > 0.05). As noted in Table 2, 30%$CO_2$/70%$H_2$ fermentation produced an acetate concentration of 3.21 ± 0.18 g/L while maximum acetate production in 50%CO/50%$H_2$ and 41%CO/18%$CO_2$/41%$H_2$ fermentation were 3.71 ± 0.13 and 2.99 ± 0.01 g/L, respectively. As a result of the same acetate amount produced between 30%$CO_2$/70%$H_2$, 50%CO/50%$H_2$, and 41%CO/18%$CO_2$/41%$H_2$ fermentation as well as the lowest biomass production in 30%$CO_2$/70%$H_2$ fermentation, the highest acetate–biomass yield ($Y_{P/X}$) was attained in 30%$CO_2$/70%$H_2$ fermentation up to 22.46 ± 0.57 g/L. This observation indicates that more carbon went to acetate synthesis than biomass formation in 30%$CO_2$/70%$H_2$ fermentation.

Other parameters assessed in this work were the correlation between the duration of the adaptation, acetate production, and volumetric productivity. A short adaptation phase and high acetate production had a positive implication on the highest acetate volumetric productivity up to 0.05 ± 0.00 g/L/h. The acetate volumetric productivity for $CO_2$/$H_2$ fermentation was even 1.64-fold higher than CO fermentation with yeast extract. Meanwhile, CO fermentation gave the lowest acetate volumetric productivity, only 0.01 ± 0.00 g/L, due to the longest adaptation phase and the lowest acetate production.

In CO fermentation, the gas was incompletely consumed, represented by a low gas pressure drop (Figure 3a). The gas analysis of the remaining syngas composition at the end of fermentation, as seen in Figure 4, showed that half of the initial CO was still left, while a small fraction of $CO_2$ and $H_2$ still remained at the end of CO fermentation. Complete gas consumption was observed in $CO_2$/$H_2$ and CO/$H_2$ fermentation. However, when $CO_2$ was added, like in CO/$CO_2$/$H_2$ fermentation, the final gas composition still contained a small amount of CO, $CO_2$, and $H_2$.

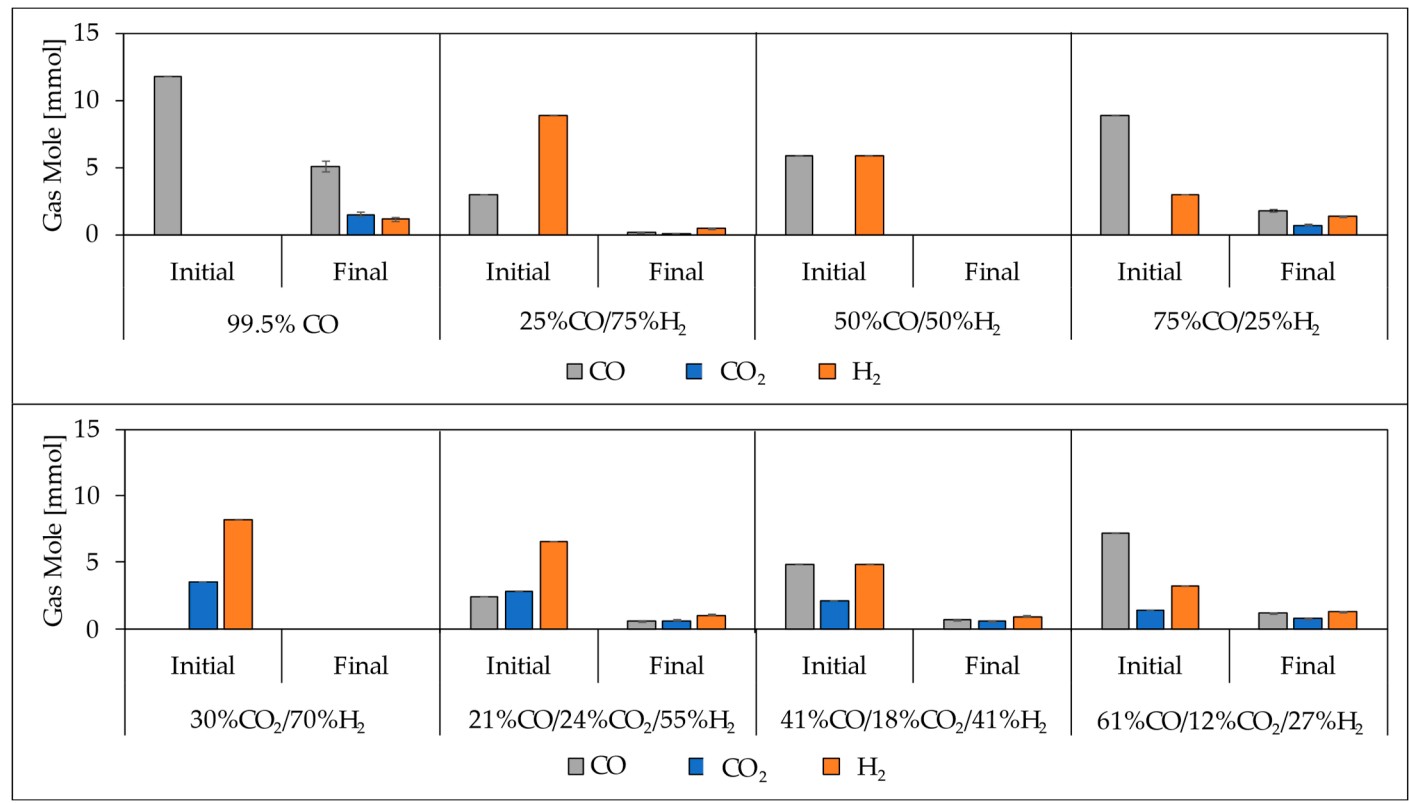

**Figure 4.** Syngas mole before and after fermentation at various syngas compositions and ratio.

### 3.3. Effect of $CO/H_2$ Ratio

Figure 5 shows that the $CO/H_2$ ratio influenced cell production, growth rate, gas consumption, and acetate formation. The cell growth rate and maximum cell production in 50%CO/50%$H_2$ declined with decreasing CO proportion in the syngas blend from 50% to 25% and increased when the proportion was increased from 50% to 75%. A lower cell growth rate and cell production in 25%CO/75%$H_2$ fermentation were associated with lower acetate production and volumetric productivity compared to 50%CO/50%$H_2$. After the fermentation using 25%CO/75%$H_2$, a tiny amount of the gas, less than 3 psi, was still present. A substantial decrease in $H_2$ showed $H_2$ utilization for converting CO fermentation-derived $CO_2$ into acetate.

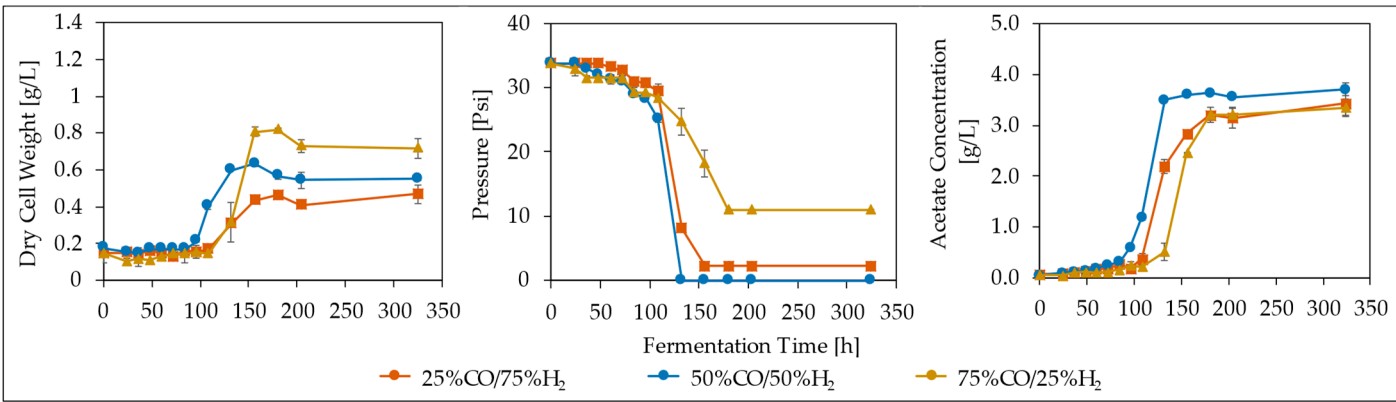

**Figure 5.** The profile of cell growth, gas pressure drops, and acetate concentration during $CO/H_2$ fermentation in the absence of yeast extract addition.

In contrast, increasing the CO proportion from 50% to 75% formed more cells instead of generating less acetate, leading to a higher $Y_{P/X}$ of $7.56 \pm 0.14$ g-acetate/g-cell for 50%CO/50%$H_2$ than 75%CO/25%$H_2$ ($4.62 \pm 0.03$ g-acetate/g-cell). Thus, the maximum volumetric productivity in 75%CO/25%$H_2$ was 1.44-fold lower than in 50%CO/50%$H_2$. When the CO percentage was increased up to 99.5%, both cell and acetate production decreased. Fermentation using 75%CO/25%$H_2$ resulted in a large amount of gas, up to 13 psi, left after the fermentation. The gas was composed of CO, $H_2$, and $CO_2$. Increasing CO to 75% promoted more CO utilization as an electron donor for *M. thermoacetica* over $H_2$. Therefore, a high $H_2$ amount still remained at the end of fermentation.

*3.4. Effect of CO/CO$_2$/H$_2$ Ratios*

As mentioned above, fermentation using $CO_2$/$H_2$ or the disappearance of CO in the syngas blend generated less biomass but resulted in the highest volumetric productivity due to high acetate production and short lag phase. We further investigated the effect of CO added to $CO_2$/$H_2$ (CO/CO$_2$/H$_2$ fermentation). Figure 6 showed that the decreasing $CO_2$/$H_2$ proportion and adding 25%CO into a gaseous mixture enhanced biomass production. In contrast, a slight increase in biomass was observed when the CO percentage was increased up to 61% in CO/CO$_2$/H$_2$ fermentation. Also, there was an insignificant increase in acetate production with increasing CO fraction in the syngas blend from 21% to 61%. Higher acetate and a slightly lower cell amount in 21%CO/24%$CO_2$/55%$H_2$ fermentation resulted in a high $Y_{X/P}$ of $6.38 \pm 0.16$ g-acetate/g-cell higher than 41%CO/18%$CO_2$/41%$H_2$ and 61%CO/12%$CO_2$/27%$H_2$, which only had $4.30 \pm 0.10$ and $4.84 \pm 0.57$ g-acetate/g-cell, respectively. By comparing 30%$CO_2$/70%$H_2$ and 21%CO/24%$CO_2$/55%$H_2$ fermentation results, it is evident that the presence of CO increased biomass production. However, the too-high CO portion had a negative impact on biomass cell and acetate production. Nonetheless, the maximum specific growth rate insignificantly increased with increasing CO fraction in CO/CO$_2$/H$_2$ fermentation ($p$-value > 0.05).

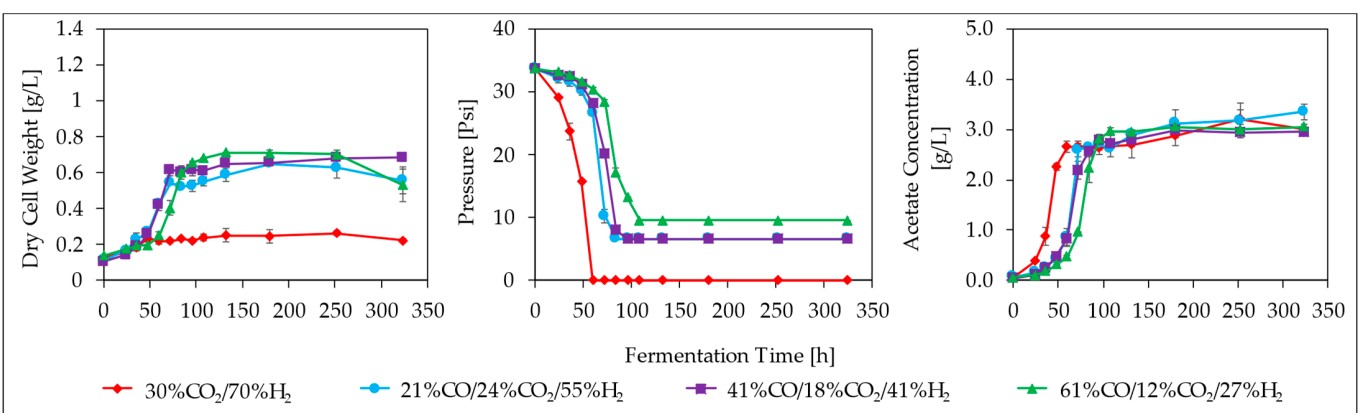

**Figure 6.** The profile of cell growth, gas pressure drops, and acetate concentration during CO/CO$_2$/H$_2$ fermentation without yeast extract addition.

Gas composition analysis (Figure 4) showed that CO, $CO_2$, and $H_2$ were still present at the end of CO/CO$_2$/H$_2$ fermentation. Like the result of the CO/$H_2$ fermentation study, in CO/CO$_2$/H$_2$ fermentation, *M. thermoacetica* consumed less $H_2$ at a higher CO proportion. On the contrary, the low CO percentage in the syngas blend promoted more $H_2$ utilization. Even though CO/CO$_2$/H$_2$ fermentation incompletely consumed the gas, the lag phase of this fermentation was shorter, and the volumetric productivity was higher than CO/$H_2$ fermentation. Fermentation using 21%CO/24%$CO_2$/55%$H_2$ showed higher acetate volumetric productivity $0.04 \pm 0.00$ g/L/h than the other syngas composition, except 30%$CO_2$/70%$H_2$ giving $0.05 \pm 0.00$ g/L/h due to the shortest lag phase.

### 3.5. Carbon Distribution and Carbon Conversion Efficiency of Acetate

Figure 7 presents the carbon distribution and carbon conversion efficiency of acetate at the initial and the end of the exponential phase for all syngas compositions. Overall, carbon from syngas was distributed more to acetate than biomass (Figure 7a), with the maximum CCE of acetate of 88% using less than 75%CO (Figure 7b). Syngas fermentation using 99.5%CO resulted in the lowest CCE of acetate (only 52.37%). When 25% $H_2$ was added and the CO portion was reduced (75%CO), the CCE of acetate increased up to 66.23%. Reducing the CO proportion up to 61%CO increased the CCE of acetate above 87%, but there was no significant increase in the CCE of acetate at less than 61%CO in the syngas mixture. This result indicated that the proportion of CO in the syngas blend significantly influenced the CCE of acetate.

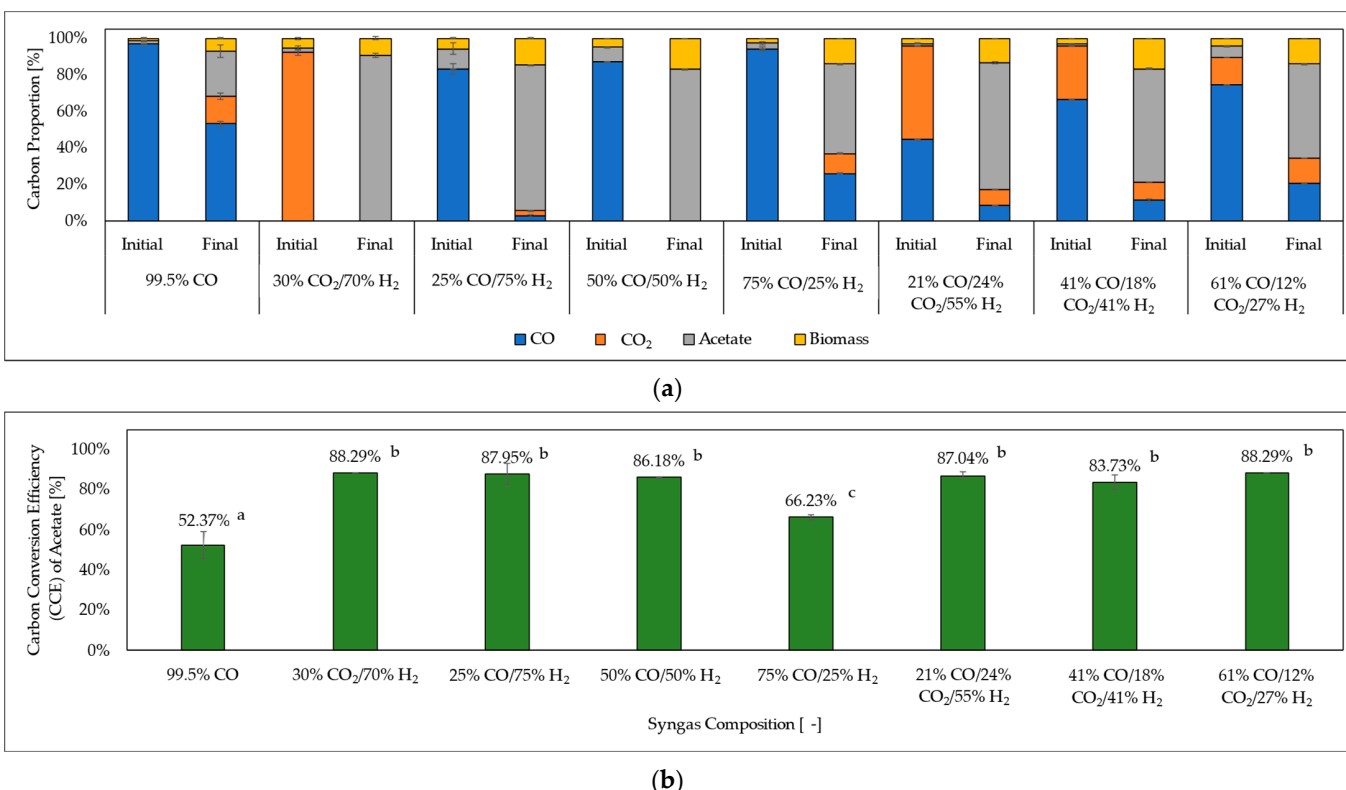

**Figure 7.** Carbon distribution at the initial and the end of the exponential phase (**a**) and carbon conversion efficiency of acetate from various syngas composition (**b**) during syngas fermentation without yeast extract addition. The different superscript letters showed that the mean among groups was significantly different (*p*-value < 0.05).

### 4. Discussion

Fermentation using CO as a sole gas of syngas blends at a high CO partial pressure negatively affected cell growth and acetate synthesis. As observed by Hurst et al., acetate production increased when the CO was increased up to 15.43 psi (1.05 atm) during CO fermentation by *C. carboxidivorans* [36]. However, inhibition occurred when the partial pressure was more than 29.39 psi (2 atm), causing low acetate production. Other studies also showed that CO partial pressure of 29.39 psi (2.0 atm) and 30.86 psi (2.1 atm) inhibited the growth of *Eubacterium limosum* KIST612 [37] and *C. carboxidivorans* P7 [24], respectively, as well as acetate production. The result of previous studies was consistent with the present study using *M. thermoacetica* fermenting 99.5%CO at 33.65 psi (2.29 atm). In this work, high CO partial pressure used for 99.5%CO fermentation caused low acetate titer, volumetric productivity, and specific growth rate and also impacted long adaptation phase duration. This is because the key enzymes playing an essential role in oxidizing CO into $CO_2$, such

as carbon monoxide dehydrogenase (CODH), get inhibited due to high concentration of CO diffusing into the cell [47]. CODH is a carbon-monoxide-sensitive enzyme containing nickel and iron clusters, allowing CO to bind to the active side of CODH via back bonding at a certain amount [13,48].

However, those parameters could be increased by adding 2 g/L yeast extract, as shown in this study. The result of the present study assessing yeast extract addition effect on CO fermentation performance further aligned with former studies [40,42], where they found that additional nutrients rich in nitrogen, vitamins, and minerals improved syngas fermentation performance. Nevertheless, Infantes et al. reported no considerable improvement in cell growth, acetate production, and syngas consumption using 0.5 g/L yeast extract during syngas fermentation by *Clostridium Ljungdalii* [49]. Infantes et al., using *C. Ljungdahlii,* found that this strain was fixing nitrogen available in the headspace into cell production instead of utilizing yeast extract ([50,51]).

Although supplementing the media with 2 g/L yeast extract could enhance acetate cell growth during the fermentation of CO, yeast extract would add significant cost to the fermentation, as shown in Table 1. The cost of yeast extract found in our study was almost similar to the additional media cost of 32% reported by Gao et al. [52]. In this study, we made a simple economic assessment based on the current acetate price ($0.09/g-sodium acetate following Sigma-Aldrich price in September 2023), the acetate production from this study (2.38 g/L and 3.85 g/L with and without yeast extract addition, respectively), and yeast extract expenses ($0.562 per 2 g/L yeast extract) without including fixed cost and other variable costs. The result showed that yeast extract addition negatively impacted the total profit. According to our calculation, the revenue of acetate production without yeast extract addition was $0.222/g-acetate, while that with 2 g/L yeast extract was $0.359/g-acetate. However, due to costly yeast extract expense accounting for $0.562/g-acetate, a substantial decrease in profit was observed to 91% of the profit from acetate production without yeast extract addition. This clearly shows that other inexpensive additional nutrients are needed for the production of acetate from syngas.

Several previous studies suggested replacing yeast extract with a nutrient that had a similar composition to yeast extract. For instance, Kundiyana et al. used cotton seed extract priced 50% cheaper than yeast extract [43]. Thi et al. compared the use of corn steep liquor (CSL), malt extract, and vegetable extract that had lower prices than yeast extract for syngas fermentation [41]. This previous study reported an appreciable enhancement in syngas fermentation performance using malt and vegetable extract instead of yeast extract. CSL, a far cheaper growth nutrient than yeast extract, did both increase the specific growth rate and acetate production [44]. Another alternative to improve acetate production is to use the optimal syngas composition, as shown in this study.

Fermentation using 50%CO/50%$H_2$, 41%CO/18%$CO_2$/41%$H_2$, and 30%$CO_2$/70%$H_2$ reached higher productivity than 99.5%CO fermentation with 2 g/L yeast extract. High productivity was achieved with high acetate production and a shorter lag phase. The observation of longer adaptation times duration CO fermentation compared to $CO_2$/$H_2$ fermentation was similar to results reported by Kerby and Zeikus using the same strain as in the present study [32]. Despite the shortest lag phase, $CO_2$/$H_2$ fermentation produced the lowest cell and specific growth rate of all the fermentations tested. As reported by Hermann using *C. Ljungdahlii* as a biocatalyst, the bacterium grew faster in CO-fermentation and was the preferred electron and carbon source for growth compared to $CO_2$/$H_2$ [53]. We did, however, find that too high of a CO concentration will lead to cell inhibition. Even though cell inhibition occurred at 99.5% CO at 2.29 atm (19 psig), a lower percentage of CO in the syngas mixture could promote cell formation and cause low $H_2$ consumption, as reported in the present study. For instance, higher cell production was observed with increasing CO proportion from 25% to 75% in CO/$H_2$ fermentation. Moreover, adding CO into $CO_2$/$H_2$ fermentation also increased cell production.

Thermodynamic analysis carried out by Hu et al. revealed that electron generation from CO was more favorable than $H_2$ [25]. This result was further described by Menon

and Ragsdale, pointing out that for the same cell mass, the CO oxidation rate by *C. thermoaceticum* was far higher than the $H_2$ uptake rate, implying that this bacterium preferred utilizing CO as the electron donor to $H_2$ [54]. Furthermore, $H_2$ has a lower solubility in water than CO, thereby requiring higher $H_2$ partial pressure or other strategies to increase $H_2$ mass transfer [55]. This makes $H_2$ a bottleneck in syngas fermentation, particularly as the electron-donation agent. More electrons are generated from CO oxidation catalyzed by CODH than $H_2$ by hydrogenase. Through a proton-gradient mechanism, ATP is further produced using available electrons from CO oxidation because no net ATP is synthesized from WLP [16]. The ATP is subsequently used for cell growth. Zhu et al. proposed a metabolic scheme of CO and $CO_2/H_2$ fermentation for *C. Ljungdahlii* to calculate ATP synthesis [56]. This scheme showed that CO fermentation synthesized more ATP than $H_2$ fermentation, influencing high-cell production. This can further explain why CO presence in syngas mixture improved biomass production. Besides the reason of more ATP synthesized from CO fermentation than from $H_2$, fermentation using 50%CO/50%$H_2$ and 41%CO/18%$CO_2$/41%$H_2$ provided more carbon than 30%$CO_2$/70%$H_2$. The excessive carbon possibly explains the larger biomass production.

However, higher biomass production after adding or increasing CO proportion in the syngas mixture was not associated with enhancing acetate production, both in CO/$H_2$ and CO/$CO_2$/$H_2$ fermentation. One of the probable reasons why more cell is produced than acetate could be a higher cell formation rate than the acetate synthesis rate in the presence of CO. While acetate is being produced, the pH of the medium drops and generates a more undissociated form of acetate. Several studies mentioned that pH below 5 and the presence of an undissociated form of acetate between 40 and 50 mM inhibited *M. thermoacetica* growth [57,58]. This can eventually lead to a decrease in acetate synthesis.

In the present study, gas analysis showed that all gases were completely consumed during the fermentation of gas compositions of 50%CO/50%$H_2$ and 30%$CO_2$/70%$H_2$. The stoichiometric analysis shows that 1/3 mol $CO_2$ was required for every 2/3 mol $H_2$ in $CO_2/H_2$ fermentation, whereas 1/2 mol CO is required for every 1/2 mol $H_2$. Interestingly, during fermentation of 41%CO/18%$CO_2$/41%$H_2$, a small amount of CO, $CO_2$, and $H_2$ remained after the fermentation. Theoretically, $CO_2$ was the only gas remaining, but both CO and $H_2$ were still present due to incomplete consumption. A possible explanation could be that $CO_2$ is inhibited during CO oxidation. Assessment of CO/$CO_2$ and CO/$CO_2$/$H_2$ ratio effect conducted by Esquivel-Elizondo et al. reported that $CO_2$ inhibited CO oxidation during syngas fermentation in mixed culture, showing more than 70% CO which was not consumed [30]. However, the $CO_2$ inhibition mechanism to CO oxidation for *M. thermoacetica* still needs studying.

Besides the $CO_2$ effect on CO oxidation, we also found that the presence of $CO_2$ together with CO in syngas caused less $H_2$ consumption. However, when one of both gases was removed, and the initial gas composition was then adjusted to 50%CO/50%$H_2$ or 30%$CO_2$/70%$H_2$, 100% gas utilization was reached. The presence of CO might inhibit the hydrogenase enzyme activity, causing low $H_2$ oxidation, as reported by Jack et al. for *C. Ljungdahlii* fermentation [26]. Syngas fermentation at an $H_2$/CO ratio ranging from 1.0 to 1.5 led to an inhibitory effect for hydrogenase.

In CO fermentation, aside from CO, $H_2$ and $CO_2$ were also detected at the end of fermentation. In WLP, CO oxidation could generate $CO_2$ and $H_2$ following the equation below:

$$CO + H_2O \rightarrow CO_2 + H_2. \tag{9}$$

Meanwhile, the high CO amount that remained in CO fermentation could be due to cell inhibition by too-high CO concentrations, ultimately impacting low syngas conversion into acetate. Unlike 99.5% fermentation, the CO proportion used in 50%CO/50%$H_2$ and 41%CO/18%$CO_2$/41%$H_2$ were found to be tolerated by *M. thermoacetica*. Therefore, the CO was almost completely consumed.

Interestingly, gas composition analysis showed that less $H_2$ was utilized at high CO proportions in CO/$H_2$ and CO/$CO_2$/$H_2$. This might be because high CO concentration

inhibited hydrogenase, impacting low $H_2$ consumption. Meanwhile, incomplete CO conversion in 75%CO/25%$H_2$ fermentation was restricted by high CO concentrations, causing a low cell consumption rate. The cell was inhibited when CO was increased from 75% to 99.5%, causing low cell and acetate production. On the other hand, the lowest cell formation and acetate production were observed in 25%CO/75%$H_2$ fermentation due to too low carbon and electron sources.

Overall, although the high gas consumption, acetate, and biomass production were achieved using syngas composition of 50%CO/50%$H_2$, the productivity was lower than $CO_2$/$H_2$ and CO/$CO_2$/$H_2$ fermentation. A CO proportion of 25% in CO/$CO_2$/$H_2$ fermentation was suggested to be used for acetate production because above that percentage led to low acetate volumetric productivity. This result also confirmed that syngas composition adjustment prior to syngas fermentation can enhance acetate production and productivity apart from costly yeast extract addition.

The syngas-to-acetate conversion efficiency using 99.5%CO demonstrated the lowest CCE of acetate ($52.37 \pm 6.77\%$), followed by syngas fermentation using 75%CO/25%$H_2$ reaching a CCE of acetate of $66.23 \pm 1.43\%$. This result corresponded to the theoretical CCE of acetate. According to Equation (1), 50% carbon from CO is utilized for $CO_2$ formation. Thus, only half of the carbon from CO is converted into acetate. However, this study showed that the rest of the carbon from syngas was not only used for $CO_2$ formation but also for biomass production, as shown in carbon distribution (Figure 7a).

Furthermore, when $H_2$ was added to the fermentation, the CCE of acetate increased. This is because $CO_2$ generated from CO fermentation was further converted into acetate. According to Equation (2), 100% carbon from $CO_2$ is distributed to acetate. The CCE of acetate increased when the $CO_2$ and $H_2$ proportion was increased, and the CO percentage was decreased. When 30%$CO_2$ and 50%CO were used for syngas fermentation, the CCE of acetate only reached $88.29 \pm 0.00\%$ and $86.18 \pm 0.17\%$, respectively, since the rest of carbon of syngas was transformed into biomass as shown in carbon distribution (Figure 7a).

The present work produced lower acetate titer than previous studies using other strains of *M. thermoacetica*. For instance, Kerby et al. found that acetate production of 4.72 g/L and 4.49 g/L in 20%$CO_2$/80%$H_2$ and 100%CO, respectively, using *M. thermoacetica* DSM 2955 (or ATCC 35608) [32]. Our study generated 3.209 g/L acetate in $CO_2$/$H_2$ fermentation in the absence of yeast extract and 3.847 g/L acetate in CO fermentation with 2 g/L yeast extract. Lower acetate production could be because the $CO_2$ used in the present study (30%$CO_2$) was higher than the previous study (20%$CO_2$), influencing the pH of the fermentation media. In addition, Kerby et al. used higher yeast extract concentration, 3 g/L yeast extract in CO fermentation and 0.5 g/L in $CO_2$/$H_2$ fermentation, while this study used 2 g/L and in the absence of yeast extract for CO and $CO_2$/$H_2$ fermentation, respectively.

The acetate production in this current study could be enhanced by using an immobilized bioreactor with more biomass and lower mass transfer limitations. Several former studies reported syngas fermentation using *M. thermoacetica* in a bioreactor. For example, Rabemanolontsoa et al. reported that a stirred tank bioreactor with continuous syngas flow at 0.4 L/min $CO_2$ using *M. thermoacetica* ATCC 39073 produced 9.3 g/L acetate [59]. However, this process supplemented 10 g/L glucose as another carbon source. Bubble column bioreactor using a genetically engineered strain of *M. thermoacetica* DSM 6867 with gas composition of 70%CO/30%$CO_2$ and 40%CO/30%$H_2$/30%$CO_2$, could produce 31 g/L and 26 g/L acetate, respectively [35]. However, these previous studies on larger-scale syngas fermentation still have not used the optimum syngas composition and have not optimized the medium cost. In further studies, we can test syngas fermentation by *M. thermoacetica* using the syngas composition suggested in this study in various bioreactor types for optimizing acetate production. By using the low-priced medium composition and the proper syngas composition with the optimum acetate production, we can enhance the economic feasibility of syngas fermentation for commercial acetate production.

## 5. Conclusions

We concluded that syngas composition and its ratios significantly influenced cell growth rate and cell production, gas consumption, and acetate formation. Syngas fermentation involving CO in blends of $CO/H_2$ and $CO/CO_2/H_2$ promoted biomass production. Higher biomass production was obtained when the proportion of CO was increased up to 75% for $CO/H_2$ fermentation and 61% for $CO/CO_2/H_2$ fermentation. However, acetate production did not improve, and the lag phase was long when the CO percentage was increased. The highest acetate volumetric productivity was achieved during $30\%CO_2/70\%H_2$ fermentation or $21\%CO/24\%CO_2/55\%H_2$ fermentation. Although a higher syngas proportion in the syngas mixture accumulated more biomass, syngas fermentation using CO as a sole gas at 19 psig generated the lowest acetate and biomass production. Acetate production could be enhanced by adding the yeast extract. It must be considered that this growth nutrient has a high cost and can add up to 39% extra cost of the fermentation medium. Accordingly, the use of yeast extract should be avoided and could be replaced with lower-priced nutrients.

**Author Contributions:** Conceptualization, B.M.H. and B.K.A.; writing—original draft preparation, B.M.H.; writing—review and editing, B.M.H. and B.K.A.; supervision, B.K.A.; project administration, B.K.A.; funding acquisition, B.K.A. All authors have read and agreed to the published version of the manuscript.

**Funding:** This work was funded by the Fulbright Scholarship to Budi Mandra Harahap and WSU CAHNRS Appendix A research program.

**Institutional Review Board Statement:** Not applicable.

**Informed Consent Statement:** Not applicable.

**Data Availability Statement:** Not applicable.

**Conflicts of Interest:** The author declares no conflict of interest.

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
