# Peer review of "Acetate Production by Moorella thermoacetica via Syngas Fermentation: Effect of Yeast Extract and Syngas Composition"

_fermentation, doi:10.3390/fermentation9090826_

Round 1
Reviewer 1 Report
The article is well presented and the study was discussed well. Minor revision is required.
1. The authors should provide clear and in-depth description of the mechanism of CO consumption (with diagram) and its effects on the fermentation, cell wall and acetate concentration.
2. Deepen the discussion and not mere result presentation backed by similar findings. What are your opinion on your results trends, sections should be concluded properly?
3. Discuss the study gap, prospects, and recommendations.
Author Response
Dear Reviewer 1,
Thank you for your good comments, which are helpful in improving our manuscript.
We attached our responses to your comments in a Word file. Please find it attached below.
Thank you

Reviewer 2 Report
This paper presents an interesting hypothesis regarding effects of yeast extract and syngas composition on the performance of acetate fermentation. Appropriate analysis methods are utilized to investigate the objectives. Although there is no innovative strategy or something new to enhance productivity, associated with a novelty, the manuscript is well organized, and the contents are rich. Therefore, I recommend the manuscript for publication after the authors address a few comments listed.
1) Is there any particular reason for testing the effect of yeast addition? According to the results of this study, its effect was significant, which means that it deserves to be introduced in the Introduction chapter.
2) If it is possible, how about suggesting an acetate conversion efficiency compared to theoretical acetate production from the consumed CO or other substrate? Showing mass balance results also can be another option.
3) Is there any possibility of producing other products such as acetone or something?
4) In the Discussion chapter, you explained the inhibition of CO with a unit of atm, which differs from your results in the figures. I suggest you use the same unit, Psi or atm, to improve readability for readers.
5) In the simple economic assessment, it is better to consider comparing the profit from the increased acetate production and the cost for yeast extract addition, not just considering the total expense.
Author Response
Dear Reviewer 2,
Thank you for your good comments, which are helpful in improving our manuscript.
We attached our responses to your comments in a Word file. Please find it attached below.
Thank you
